# Stochastic Gradient MCMC with Stale Gradients

**Changyou Chen**[†]      **Nan Ding**[‡]      **Chunyuan Li**[†]      **Yizhe Zhang**[†]      **Lawrence Carin**[†]

[†]Dept. of Electrical and Computer Engineering, Duke University, Durham, NC, USA

[‡]Google Inc., Venice, CA, USA

[†]`{cc448,cl319,yz196,lcarin}@duke.edu;` [‡]`dingnan@google.com`

## Abstract

Stochastic gradient MCMC (SG-MCMC) has played an important role in large-scale Bayesian learning, with well-developed theoretical convergence properties. In such applications of SG-MCMC, it is becoming increasingly popular to employ distributed systems, where stochastic gradients are computed based on some outdated parameters, yielding what are termed *stale gradients*. While stale gradients could be directly used in SG-MCMC, their impact on convergence properties has not been well studied. In this paper we develop theory to show that while the bias and MSE of an SG-MCMC algorithm depend on the staleness of stochastic gradients, its estimation variance (relative to the *expected* estimate, based on a prescribed number of samples) is independent of it. In a simple Bayesian distributed system with SG-MCMC, where stale gradients are computed asynchronously by a set of workers, our theory indicates a linear speedup on the decrease of estimation variance w.r.t. the number of workers. Experiments on synthetic data and deep neural networks validate our theory, demonstrating the effectiveness and scalability of SG-MCMC with stale gradients.

## 1   Introduction

The pervasiveness of big data has made scalable machine learning increasingly important, especially for deep models. A basic technique is to adopt stochastic optimization algorithms [1], *e.g.*, stochastic gradient descent and its extensions [2]. In each iteration of stochastic optimization, a minibatch of data is used to evaluate the gradients of the objective function and update model parameters (errors are introduced in the gradients, because they are computed based on minibatches rather than the entire dataset; since the minibatches are typically selected at random, this yields the term "stochastic" gradient). This is highly scalable because processing a minibatch of data in each iteration is relatively cheap compared to analyzing the entire (large) dataset at once. Under certain conditions, stochastic optimization is guaranteed to converge to a (local) optima [1]. Because of its scalability, the minibatch strategy has recently been extended to Markov Chain Monte Carlo (MCMC) Bayesian sampling methods, yielding SG-MCMC [3, 4, 5].

In order to handle large-scale data, distributed stochastic optimization algorithms have been developed, for example [6], to further improve scalability. In a distributed setting, a cluster of machines with multiple cores cooperate with each other, typically through an asynchronous scheme, for scalability [7, 8, 9]. A downside of an asynchronous implementation is that stale gradients must be used in parameter updates ("stale gradients" are stochastic gradients computed based on outdated parameters, instead of the latest parameters; they are easier to compute in a distributed system, but introduce additional errors relative to traditional stochastic gradients). While some theory has been developed to guarantee the convergence of stochastic optimization with stale gradients [10, 11, 12], little analysis has been done in a Bayesian setting, where SG-MCMC is applied. Distributed SG-MCMC algorithms share characteristics with distributed stochastic optimization, and thus are highly scalable and suitable for large-scale Bayesian learning. Existing Bayesian distributed systems with traditional MCMC methods, such as [13], usually employ stale *statistics* instead of stale *gradients*, where stale statistics

are summarized based on outdated parameters, *e.g.*, outdated topic distributions in distributed Gibbs sampling [13]. Little theory exists to guarantee the convergence of such methods. For existing distributed SG-MCMC methods, typically only standard stochastic gradients are used, for limited problems such as matrix factorization, without rigorous convergence theory [14, 15, 16].

In this paper, by extending techniques from standard SG-MCMC [17], we develop theory to study the convergence behavior of SG-MCMC with Stale gradients (S²G-MCMC). Our goal is to evaluate the *posterior average* of a test function $\phi(\mathbf{x})$, defined as $\bar{\phi} \triangleq \int_{\mathcal{X}} \phi(\mathbf{x})\rho(\mathbf{x})\mathrm{d}\,\mathbf{x}$, where $\rho(\mathbf{x})$ is the desired posterior distribution with $\mathbf{x}$ the possibly augmented model parameters (see Section 2). In practice, S²G-MCMC generates $L$ samples $\{\mathbf{x}_l\}_{l=1}^{L}$ and uses the *sample average* $\hat{\phi}_L \triangleq \frac{1}{L}\sum_{l=1}^{L}\phi(\mathbf{x}_l)$ to approximate $\bar{\phi}$. We measure how $\hat{\phi}_L$ approximates $\bar{\phi}$ in terms of *bias*, *MSE* and *estimation variance*, defined as $|\mathbb{E}\hat{\phi}_L - \bar{\phi}|$, $\mathbb{E}\left(\hat{\phi}_L - \bar{\phi}\right)^2$ and $\mathbb{E}\left(\hat{\phi}_L - \mathbb{E}\hat{\phi}_L\right)^2$, respectively. From the definitions, the bias and MSE characterize how accurately $\hat{\phi}_L$ approximates $\bar{\phi}$, and the variance characterizes how fast $\hat{\phi}_L$ converges to its own expectation (for a prescribed number of samples $L$). Our theoretical results show that while the bias and MSE depend on the staleness of stochastic gradients, the variance is independent of it. In a simple *asynchronous Bayesian distributed system* with S²G-MCMC, our theory indicates a linear speedup on the decrease of the variance w.r.t. the number of workers used to calculate the stale gradients, while *maintaining the same optimal bias level as standard SG-MCMC*. We validate our theory on several synthetic experiments and deep neural network models, demonstrating the effectiveness and scalability of the proposed S²G-MCMC framework.

**Related Work**  Using stale gradients is a standard setup in distributed stochastic optimization systems. Representative algorithms include, but are not limited to, the ASYSG-CON [6] and HOG-WILD! algorithms [18], and some more recent developments [19, 20]. Furthermore, recent research on stochastic optimization has been extended to non-convex problems with provable convergence rates [12]. In Bayesian learning with MCMC, existing work has focused on running parallel chains on subsets of data [21, 22, 23, 24], and little if any effort has been made to use stale stochastic gradients, the setting considered in this paper.

## 2   Stochastic Gradient MCMC

Throughout this paper, we denote vectors as bold lower-case letters, and matrices as bold upper-case letters. For example, $\mathcal{N}(\mathbf{m}, \boldsymbol{\Sigma})$ means a multivariate Gaussian distribution with mean $\mathbf{m}$ and covariance $\boldsymbol{\Sigma}$. In the analysis we consider algorithms with fixed-stepsizes for simplicity; decreasing-stepsize variants can be addressed similarly as in [17].

The goal of SG-MCMC is to generate random samples from a posterior distribution $p(\boldsymbol{\theta}|\,\mathbf{D}) \propto p(\boldsymbol{\theta})\prod_{i=1}^{N}p(\mathbf{d}_i\,|\,\boldsymbol{\theta})$, which are used to evaluate a test function. Here $\boldsymbol{\theta} \in \mathbb{R}^n$ represents the parameter vector and $\mathbf{D} = \{\mathbf{d}_1, \cdots, \mathbf{d}_N\}$ represents the data, $p(\boldsymbol{\theta})$ is the prior distribution, and $p(\mathbf{d}_i\,|\,\boldsymbol{\theta})$ the likelihood for $\mathbf{d}_i$. SG-MCMC algorithms are based on a class of stochastic differential equations, called Itô diffusion, defined as

$$\mathrm{d}\,\mathbf{x}_t = F(\mathbf{x}_t)\mathrm{d}t + g(\mathbf{x}_t)\mathrm{d}\mathbf{w}_t \,, \tag{1}$$

where $\mathbf{x} \in \mathbb{R}^m$ represents the model states, typically $\mathbf{x}$ augments $\boldsymbol{\theta}$ such that $\boldsymbol{\theta} \subseteq \mathbf{x}$ and $n \leq m$; $t$ is the time index, $\mathbf{w}_t \in \mathbb{R}^m$ is $m$-dimensional Brownian motion, functions $F : \mathbb{R}^m \to \mathbb{R}^m$ and $g : \mathbb{R}^m \to \mathbb{R}^{m \times m}$ are assumed to satisfy the usual Lipschitz continuity condition [25].

For appropriate functions $F$ and $g$, the stationary distribution, $\rho(\mathbf{x})$, of the Itô diffusion (1) has a marginal distribution equal to the posterior distribution $p(\boldsymbol{\theta}|\,\mathbf{D})$ [26]. For example, denoting the unnormalized negative log-posterior as $U(\boldsymbol{\theta}) \triangleq -\log p(\boldsymbol{\theta}) - \sum_{i=1}^{N}\log p(\mathbf{d}_i\,|\,\boldsymbol{\theta})$, the stochastic gradient Langevin dynamic (SGLD) method [3] is based on 1st-order Langevin dynamics, with $\mathbf{x} = \boldsymbol{\theta}$, and $F(\mathbf{x}_t) = -\nabla_{\boldsymbol{\theta}}U(\boldsymbol{\theta})$, $g(\mathbf{x}_t) = \sqrt{2}\,\mathbf{I}_n$, where $\mathbf{I}_n$ is the $n \times n$ identity matrix. The stochastic gradient Hamiltonian Monte Carlo (SGHMC) method [4] is based on 2nd-order Langevin dynamics, with $\mathbf{x} = (\boldsymbol{\theta}, \mathbf{q})$, and $F(\mathbf{x}_t) = \begin{pmatrix} \mathbf{q} \\ -B\,\mathbf{q} - \nabla_{\boldsymbol{\theta}}U(\boldsymbol{\theta}) \end{pmatrix}, g(\mathbf{x}_t) = \sqrt{2B}\begin{pmatrix} \mathbf{0} & \mathbf{0} \\ \mathbf{0} & \mathbf{I}_n \end{pmatrix}$ for a scalar $B > 0$; $\mathbf{q}$ is an auxiliary variable known as the momentum [4, 5]. Diffusion forms for other SG-MCMC algorithms, such as the stochastic gradient thermostat [5] and variants with Riemannian information geometry [27, 26, 28], are defined similarly.

In order to efficiently draw samples from the continuous-time diffusion (1), SG-MCMC algorithms typically apply two approximations: *i*) Instead of analytically integrating infinitesimal increments

$dt$, numerical integration over small step size $h$ is used to approximate the integration of the true dynamics. *ii*) Instead of working with the full gradient $\nabla_{\boldsymbol{\theta}} U(\boldsymbol{\theta}_{lh})$, a stochastic gradient $\nabla_{\boldsymbol{\theta}} \tilde{U}_l(\boldsymbol{\theta}_{lh})$, defined as

$$\nabla_{\boldsymbol{\theta}} \tilde{U}_l(\boldsymbol{\theta}) \triangleq -\nabla_{\boldsymbol{\theta}} \log p(\boldsymbol{\theta}) - \frac{N}{J} \sum_{i=1}^{J} \nabla_{\boldsymbol{\theta}} \log p(\mathbf{d}_{\pi_i} | \boldsymbol{\theta}), \qquad (2)$$

is calculated from a minibatch of size $J$, where $\{\pi_1, \cdots, \pi_J\}$ is a random subset of $\{1, \cdots, N\}$. Note that to match the time index $t$ in (1), parameters have been and will be indexed by "$lh$" in the $l$-th iteration.

# 3 Stochastic Gradient MCMC with Stale Gradients

In this section, we extend SG-MCMC to the stale-gradient setting, commonly met in asynchronous distributed systems [7, 8, 9], and develop theory to analyze convergence properties.

## 3.1 Stale stochastic gradient MCMC (S²G-MCMC)

The setting for S²G-MCMC is the same as the standard SG-MCMC described above, except that the stochastic gradient (2) is replaced with a stochastic gradient evaluated with outdated parameter $\boldsymbol{\theta}_{(l-\tau_l)h}$ instead of the latest version $\boldsymbol{\theta}_{lh}$ (see Appendix A for an example):

$$\nabla_{\boldsymbol{\theta}} \hat{U}_{\tau_l}(\boldsymbol{\theta}) \triangleq -\nabla_{\boldsymbol{\theta}} \log p(\boldsymbol{\theta}_{(l-\tau_l)h}) - \frac{N}{J} \sum_{i=1}^{J} \nabla_{\boldsymbol{\theta}} \log p(\mathbf{d}_{\pi_i} | \boldsymbol{\theta}_{(l-\tau_l)h}), \qquad (3)$$

where $\tau_l \in \mathbb{Z}^+$ denotes the *staleness* of the parameter used to calculate the stochastic gradient in the $l$-th iteration. A distinctive difference between S²G-MCMC and SG-MCMC is that stale stochastic gradients are no longer unbiased estimations of the true gradients. This leads to additional challenges in developing convergence bounds, one of the main contributions of this paper.

We assume a bounded staleness for all $\tau_l$'s, *i.e.*,

$$\max_l \tau_l \leq \tau$$

for some constant $\tau$. As an example, Algorithm 1 describes the update rule of the stale-SGHMC in each iteration with the Euler integrator, where the stale gradient $\nabla_{\boldsymbol{\theta}} \hat{U}_{\tau_l}(\boldsymbol{\theta})$ with staleness $\tau_l$ is used.

---

**Algorithm 1** State update of SGHMC with the stale stochastic gradient $\nabla_{\boldsymbol{\theta}} \hat{U}_{\tau_l}(\boldsymbol{\theta})$

---

**Input:** $\mathbf{x}_{lh} = (\boldsymbol{\theta}_{lh}, \mathbf{q}_{lh}), \nabla_{\boldsymbol{\theta}} \hat{U}_{\tau_l}(\boldsymbol{\theta}), \tau_l, \tau, h, B$
**Output:** $\mathbf{x}_{(l+1)h} = (\boldsymbol{\theta}_{(l+1)h}, \mathbf{q}_{(l+1)h})$
**if** $\tau_l \leq \tau$ **then**
    Draw $\boldsymbol{\zeta}_l \sim \mathcal{N}(0, \mathbf{I})$;
    $\mathbf{q}_{(l+1)h} = (1 - Bh)\,\mathbf{q}_{lh} - \nabla_{\boldsymbol{\theta}} \hat{U}_{\tau_l}(\boldsymbol{\theta})h + \sqrt{2Bh}\boldsymbol{\zeta}_l$;
    $\boldsymbol{\theta}_{(l+1)h} = \boldsymbol{\theta}_{lh} + \mathbf{q}_{(l+1)h}\,h$;
**end if**

---

## 3.2 Convergence analysis

This section analyzes the convergence properties of the basic S²G-MCMC; an extension with multiple chains is discussed in Section 3.3. It is shown that the bias and MSE depend on the staleness parameter $\tau$, while the variance is independent of it, yielding significant speedup in Bayesian distributed systems.

**Bias and MSE**   In [17], the bias and MSE of the standard SG-MCMC algorithms with a $K$th order integrator were analyzed, where the order of an integrator reflects how accurately an SG-MCMC algorithm approximates the corresponding continuous diffusion. Specifically, if evolving $\mathbf{x}_t$ with a numerical integrator using discrete time increment $h$ induces an error bounded by $O(h^K)$, the integrator is called a $K$th order integrator, *e.g.*, the popular Euler method used in SGLD [3] is a 1st-order integrator. In particular, [17] proved the bounds stated in Lemma 1.

**Lemma 1** ([17]). *Under standard assumptions (see Appendix B), the bias and MSE of SG-MCMC with a $K$th-order integrator at time $T = hL$ are bounded as:*

$$\textit{Bias: } \left| \mathbb{E}\hat{\phi}_L - \bar{\phi} \right| = O\left( \frac{\sum_l \|\mathbb{E}\Delta V_l\|}{L} + \frac{1}{Lh} + h^K \right)$$

$$\textit{MSE: } \mathbb{E}\left( \hat{\phi}_L - \bar{\phi} \right)^2 = O\left( \frac{\frac{1}{L}\sum_l \mathbb{E}\|\Delta V_l\|^2}{L} + \frac{1}{Lh} + h^{2K} \right)$$

Here $\Delta V_l \triangleq \mathcal{L} - \tilde{\mathcal{L}}_l$, where $\mathcal{L}$ is the generator of the Itô diffusion (1) defined as

$$\mathcal{L}f(\mathbf{x}_t) \triangleq \lim_{h \to 0^+} \frac{\mathbb{E}\left[ f(\mathbf{x}_{t+h}) \right] - f(\mathbf{x}_t)}{h} = \left( F(\mathbf{x}_t) \cdot \nabla_{\mathbf{x}} + \frac{1}{2} \left( g(\mathbf{x}_t)g(\mathbf{x}_t)^T \right) : \nabla_{\mathbf{x}}\nabla_{\mathbf{x}}^T \right) f(\mathbf{x}_t), \quad (4)$$

for any compactly supported twice differentiable function $f : \mathbb{R}^n \to \mathbb{R}$, $h \to 0^+$ means $h$ approaches zero along the positive real axis. $\tilde{\mathcal{L}}_l$ is the same as $\mathcal{L}$ except using the stochastic gradient $\nabla \tilde{U}_l$ instead of the full gradient.

We show that the bounds of the bias and MSE of $S^2$G-MCMC share similar forms as SG-MCMC, but with additional dependence on the staleness parameter. In addition to the assumptions in SG-MCMC [17] (see details in Appendix B), the following additional assumption is imposed.

**Assumption 1.** *The noise in the stochastic gradients is well-behaved, such that: 1) the stochastic gradient is unbiased, i.e., $\nabla_{\boldsymbol{\theta}} U(\boldsymbol{\theta}) = \mathbb{E}_\xi \nabla_{\boldsymbol{\theta}} \tilde{U}(\boldsymbol{\theta})$ where $\xi$ denotes the random permutation over*

$\{1, \cdots, N\}$; *2) the variance of stochastic gradient is bounded, i.e., $\mathbb{E}_\xi \left\| U(\boldsymbol{\theta}) - \tilde{U}(\boldsymbol{\theta}) \right\|^2 \le \sigma^2$; 3) the*

*gradient function $\nabla_{\boldsymbol{\theta}} U$ is Lipschitz (so is $\nabla_{\boldsymbol{\theta}} \tilde{U}$), i.e., $\|\nabla_{\boldsymbol{\theta}} U(\mathbf{x}) - \nabla_{\boldsymbol{\theta}} U(\mathbf{y})\| \le C \|\mathbf{x} - \mathbf{y}\|, \forall \mathbf{x}, \mathbf{y}$.*

In the following theorems, we omit the assumption statement for conciseness. Due to the staleness of the stochastic gradients, the term $\Delta V_l$ in $S^2$G-MCMC is equal to $\mathcal{L} - \tilde{\mathcal{L}}_{l-\tau_l}$, where $\tilde{\mathcal{L}}_{l-\tau_l}$ arises from $\nabla_{\boldsymbol{\theta}} \hat{U}_{\tau_l}$. The challenge arises to bound these terms involving $\Delta V_l$. To this end, define $f_{lh} \triangleq \left\| \mathbf{x}_{lh} - \mathbf{x}_{(l-1)h} \right\|$, and $\psi$ to be a functional satisfying the *Poisson Equation**:

$$\frac{1}{L} \sum_{l=1}^{L} \mathcal{L}\psi(\mathbf{x}_{lh}) = \hat{\phi}_L - \bar{\phi} . \tag{5}$$

**Theorem 2.** *After $L$ iterations, the bias of $S^2$G-MCMC with a $K$th-order integrator is bounded, for some constant $D_1$ independent of $\{L, h, \tau\}$, as:*

$$\left| \mathbb{E}\hat{\phi}_L - \bar{\phi} \right| \le D_1 \left( \frac{1}{Lh} + M_1 \tau h + M_2 h^K \right) ,$$

*where $M_1 \triangleq \max_l |\mathcal{L}f_{lh}| \max_l \|\mathbb{E}\nabla\psi(\mathbf{x}_{lh})\| C$, $M_2 \triangleq \sum_{k=1}^{K} \frac{\sum_l \mathbb{E}\tilde{\mathcal{L}}_l^{k+1}\psi(\mathbf{x}_{(l-1)h})}{(k+1)!L}$ are constants.*

**Theorem 3.** *After $L$ iterations, the MSE of $S^2$G-MCMC with a $K$th-order integrator is bounded, for some constant $D_2$ independent of $\{L, h, \tau\}$, as:*

$$\mathbb{E}\left( \hat{\phi}_L - \bar{\phi} \right)^2 \le D_2 \left( \frac{1}{Lh} + \tilde{M}_1 \tau^2 h^2 + \tilde{M}_2 h^{2K} \right) ,$$

*where constants $\tilde{M}_1 \triangleq \max_l \|\mathbb{E}\nabla\psi(\mathbf{x}_{lh})\|^2 \max_l (\mathcal{L}f_{lh})^2 C^2$, $\tilde{M}_2 \triangleq \mathbb{E}(\frac{\sum_l \tilde{\mathcal{L}}_l^{K+1}\psi(\mathbf{x}_{(l-1)h})}{L(K+1)!})^2$.*
The theorems indicate that both the bias and MSE depend on the staleness parameter $\tau$. For a fixed computational time, this could possibly lead to unimproved bounds, compared to standard SG-MCMC, when $\tau$ is too large, *i.e.*, the terms with $\tau$ would dominate, as is the case in the distributed system discussed in Section 4. Nevertheless, better bounds than standard SG-MCMC could be obtained if the decrease of $\frac{1}{Lh}$ is faster than the increase of the staleness in a distributed system.

**Variance** Next we investigate the convergence behavior of the variance, $\mathrm{Var}(\hat{\phi}_L) \triangleq \mathbb{E}\left( \hat{\phi}_L - \mathbb{E}\hat{\phi}_L \right)^2$. Theorem 4 indicates the variance is independent of $\tau$, hence a linear speedup in the decrease of variance is always achievable when stale gradients are computed in parallel. An example is discussed in the Bayesian distributed system in Section 4.

**Theorem 4.** *After $L$ iterations, the variance of $S^2$G-MCMC with a $K$th-order integrator is bounded, for some constant $D$, as:*

$$\mathrm{Var}\left( \hat{\phi}_L \right) \le D \left( \frac{1}{Lh} + h^{2K} \right) .$$

The variance bound is the same as for standard SG-MCMC, whereas $L$ could increase linearly w.r.t. the number of workers in a distributed setting, yielding significant variance reduction. When optimizing the the variance bound w.r.t. $h$, we get an optimal variance bound stated in Corollary 5.

**Corollary 5.** *In term of estimation variance, the optimal convergence rate of $S^2$G-MCMC with a $K$th-order integrator is bounded as: $\mathrm{Var}\left( \hat{\phi}_L \right) \le O\left( L^{-2K/(2K+1)} \right)$.*

In real distributed systems, the decrease of $1/Lh$ and increase of $\tau$, in the bias and MSE bounds, would typically cancel, leading to the same bias and MSE level compared to standard SG-MCMC, whereas a linear speedup on the decrease of variance w.r.t. the number of workers is always achievable. More details are discussed in Section 4.

### 3.3 Extension to multiple parallel chains

This section extends the theory to the setting with $S$ parallel chains, each independently running an $S^2$G-MCMC algorithm. After generating samples from the $S$ chains, an aggregation step is needed to combine the sample average from each chain, *i.e.*, $\{\hat{\phi}_{L_s}\}_{s=1}^{M}$, where $L_s$ is the number of iterations on chain $s$. For generality, we allow each chain to have different step sizes, *e.g.*, $(h_s)_{s=1}^{S}$. We aggregate the sample averages as $\hat{\phi}_L^S \triangleq \sum_{s=1}^{S} \frac{T_s}{T} \hat{\phi}_{L_s}$, where $T_s \triangleq L_s h_s$, $T \triangleq \sum_{s=1}^{S} T_s$.

Interestingly, with increasing $S$, using multiple chains does not seem to directly improve the convergence rate for the bias, but improves the MSE bound, as stated in Theorem 6.

**Theorem 6.** *Let $T_m \triangleq \max_l T_l$, $h_m \triangleq \max_l h_l$, $\bar{T} = T/S$, the bias and MSE of $S$ parallel $S^2$G-MCMC chains with a $K$th-order integrator are bounded, for some constants $D_1$ and $D_2$ independent of $\{L, h, \tau\}$, as:*

$$\text{Bias: } \left| \mathbb{E}\hat{\phi}_L^S - \bar{\phi} \right| \leq D_1 \left( \frac{1}{\bar{T}} + \frac{T_m}{\bar{T}} \left( M_1 \tau h_s + M_2 h_s^K \right) \right)$$

$$\text{MSE: } \mathbb{E}\left( \hat{\phi}_L^S - \bar{\phi} \right)^2 \leq D_2 \left( \frac{1 - 1/\bar{T}}{T} + \frac{1}{\bar{T}^2} + \frac{T_m^2}{\bar{T}^2} \left( M_1^2 \tau^2 h_s^2 + M_2^2 h_s^{2K} \right) \right).$$

Assume that $\bar{T} = T/S$ is independent of the number of chains. As a result, using multiple chains does not directly improve the bound for the bias[†]. However, for the MSE bound, although the last two terms are independent of $S$, the first term decreases linearly with respect to $S$ because $T = \bar{T}S$. This indicates a decreased estimation variance with more chains. This matches the intuition because more samples can be obtained with more chains in a given amount of time.

The decrease of MSE for multiple-chain is due to the decrease of the variance as stated in Theorem 7.

**Theorem 7.** *The variance of $S$ parallel $S^2$G-MCMC chains with a $K$th-order integrator is bounded, for some constant $D$ independent of $\{L, h, \tau\}$, as:*

$$\mathbb{E}\left( \hat{\phi}_L^S - \mathbb{E}\hat{\phi}_L^S \right)^2 \leq D \left( \frac{1}{T} + \sum_{s=1}^{S} \frac{T_s^2}{T^2} h_s^{2K} \right).$$

When using the same step size for all chains, Theorem 7 gives an optimal variance bound of $O\left( (\sum_s L_s)^{-2K/(2K+1)} \right)$, *i.e.* a linear speedup with respect to $S$ is achieved.

In addition, Theorem 6 with $\tau = 0$ and $K = 1$ provides convergence rates for the distributed SGLD algorithm in [14], *i.e.*, improved MSE and variance bounds compared to the single-server SGLD.

## 4 Applications to Distributed SG-MCMC Systems

Our theory for $S^2$G-MCMC is general, serving as a basic analytic tool for distributed SG-MCMC systems. We propose two simple Bayesian distributed systems with $S^2$G-MCMC in the following.

**Single-chain distributed SG-MCMC**   Perhaps the simplest architecture is an asynchronous distributed SG-MCMC system, where a server runs an $S^2$G-MCMC algorithm, with stale gradients computed asynchronously from $W$ workers. The detailed operations of the server and workers are described in Appendix A.

With our theory, now we explain the convergence property of this simple distributed system with SG-MCMC, *i.e.*, a linear speedup w.r.t. the number of workers on the decrease of variance, while maintaining the same bias level. To this end, rewrite $L = W\bar{L}$ from Theorems 2 and 3, where $\bar{L}$ is the average number of iterations on each worker. We can observe from the theorems that when $M_1 \tau h > M_2 h^K$ in the bias and $\tilde{M}_1 \tau^2 h^2 > \tilde{M}_2 h^{2K}$ in the MSE, the terms with $\tau$ dominate. Optimizing the bounds with respect to $h$ yields a bound of $O((\tau/W\bar{L})^{1/2})$ for the bias, and $O((\tau/W\bar{L})^{2/3})$ for the MSE. In practice, we usually observe $\tau \approx W$, making $W$ in the optimal bounds cancels, *i.e.*, the same optimal bias and MSE bounds as standard SG-MCMC are obtained, no theoretical speedup is

---

[†]It means the bound does not directly relate to low-order terms of $S$, though constants might be improved.

achieved when increasing $W$. However, from Corollary 5, the variance is independent of $\tau$, thus a linear speedup on the variance bound can be always obtained when increasing the number of workers, *i.e.*, the distributed SG-MCMC system convergences a factor of $W$ faster than standard SG-MCMC with a single machine. We are not aware of similar conclusions from optimization, because most of the research focuses on the convex setting, thus only variance (equivalent to MSE) is studied.

**Multiple-chain distributed SG-MCMC**  We can also adopt multiple servers based on the multiple-chain setup in Section 3.3, where each chain corresponds to one server. The detailed architecture is described in Appendix A. This architecture trades off communication cost with convergence rates. As indicated by Theorems 6 and 7, the MSE and variance bounds can be improved with more servers. Note that when only one worker is associated with one server, we recover the setting of $S$ independent servers. Compared to the single-server architecture described above with $S$ workers, from Theorems 2–7, while the variance bound is the same, the single-server arthitecture improves the bias and MSE bounds by a factor of $S$.

**More advanced architectures**  More complex architectures could also be designed to reduce communication cost, for example, by extending the downpour [7] and elastic SGD [29] architectures to the SG-MCMC setting. Their convergence properties can also be analyzed with our theory since they are essentially using stale gradients. We leave the detailed analysis for future work.

# 5   Experiments

Our primal goal is to validate the theory, comparing with different distributed architectures and algorithms, such as [30, 31], is beyond the scope of this paper. We first use two synthetic experiments to validate the theory, then apply the distributed architecture described in Section 4 for Bayesian deep learning. To quantitatively describe the speedup property, we adopt the the *iteration speedup* [12], defined as: *iteration speedup* $\triangleq \frac{\text{\#iterations with a single worker}}{\text{average \#iterations on a worker}}$, where # is the iteration count when the same level of precision is achieved. This speedup best matches with the theory. We also consider the *time speedup*, defined as: $\frac{\text{running time for a single worker}}{\text{running time for } W \text{ worker}}$, where the running time is recorded at the same accuracy. It is affected significantly by hardware, thus is not accurately consistent with the theory.

## 5.1   Synthetic experiments

**Impact of stale gradients**  A simple Gaussian model is used to verify the impact of stale gradients on the convergence accuracy, with $d_i \sim \mathcal{N}(\theta, 1), \theta \sim \mathcal{N}(0, 1)$. 1000 data samples $\{d_i\}$ are generated, with minibatches of size 10 to calculate stochastic gradients. The test function is $\phi(\theta) \triangleq \theta^2$. The distributed SGLD algorithm is adopted in this experiment. We aim to verify that the optimal MSE bound $\propto \tau^{2/3} L^{-2/3}$, derived from Theorem 3 and discussed in Section 4 (with $W = 1$). The optimal stepsize is $h = C\tau^{-2/3}L^{-1/3}$ for some constant $C$. Based on the optimal bound, setting $L = L_0 \times \tau$ for some fixed $L_0$ and varying $\tau$'s would result in the same MSE, which is $\propto L_0^{-2/3}$. In the experiments we set $C = 1/30$,

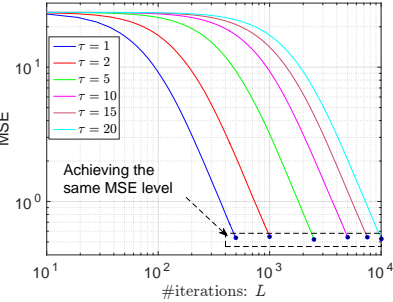

Figure 1: MSE vs. # iterations ($L = 500 \times \tau$) with increasing staleness $\tau$. Resulting in roughly the same MSE.

$L_0 = 500, \tau = \{1, 2, 5, 10, 15, 20\}$, and average over 200 runs to approximate the expectations in the MSE formula. As indicated in Figure 1, approximately the same MSE's are obtained after $L_0\tau$ iterations for different $\tau$ values, consistent with the theory. Note since the stepsizes are set to make end points of the curves reach the optimal MSE's, the curves would not match the optimal MSE curves of $\tau^{2/3}L^{-2/3}$ in general, except for the end points, i.e., they are lower bounded by $\tau^{2/3}L^{-2/3}$.

**Convergence speedup of the variance**  A Bayesian logistic regression model (BLR) is adopted to verify the variance convergence properties. We use the Adult dataset[‡], a9a, with 32,561 training samples and 16,281 test samples. The test function is defined as the standard logistic loss. We average over 10 runs to estimate the expectation $\mathbb{E}\hat{\phi}_L$ in the variance. We use the single-server distributed architecture in Section 4, with multiple workers computing stale gradients in parallel. We plot the variance versus the average number of iterations on the workers ($\bar{L}$) and the running time in Figure 2 (a) and (b), respectively. We can see that the variance drops faster with increasing number

---

[‡]http://www.csie.ntu.edu.tw/ cjlin/libsvmtools/datasets/binary.html.

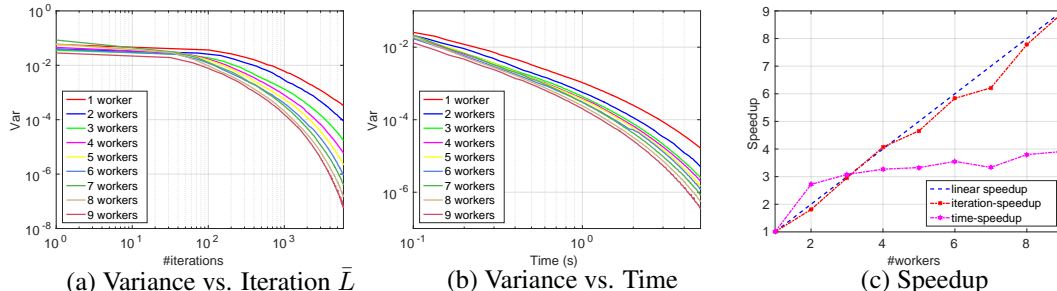

(a) Variance vs. Iteration $\bar{L}$      (b) Variance vs. Time      (c) Speedup

Figure 2: Variance with increasing number of workers.

of workers. To quantitatively relate these results to the theory, Corollary 5 indicates that $\frac{L_1}{L_2} = \frac{W_1}{W_2}$, where $(W_i, L_i)_{i=1}^2$ means the number of workers and iterations at the same variance, *i.e.*, a linear speedup is achieved. The *iteration speedup* and *time speedup* are plotted in Figure 2 (c), showing that the *iteration speedup* approximately scales linearly worker numbers, consistent with Corollary 5; whereas the *time speedup* deteriorates when the worker number is large due to high system latency.

## 5.2 Applications to deep learning

We further test S$^2$G-MCMC on Bayesian learning of deep neural networks. The distributed system is developed based on an MPI (message passing interface) extension of the popular Caffe package for deep learning [32]. We implement the SGHMC algorithm, with the point-to-point communications between servers and workers handled by the MPICH library. The algorithm is run on a cluster of five machines. Each machine is equipped with eight 3.60GHz Intel(R) Core(TM) i7-4790 CPU cores.

We evaluate S$^2$G-MCMC on the above BLR model and two deep convolutional neural networks (CNN). In all these models, zero mean and unit variance Gaussian priors are employed for the weights to capture weight uncertainties, an effective way to deal with overfitting [33]. We vary the number of servers $S$ among $\{1, 3, 5, 7\}$, and the number of workers for each server from 1 to 9.

**LeNet for MNIST** We modify the standard LeNet to a Bayesian setting for the MNIST dataset. LeNet consists of 2 convolutional layers, 2 max pool layers and 2 ReLU nonlinear layers, followed by 2 fully connected layers [34]. The detailed specification can be found in Caffe. For simplicity, we use the default parameter setting specified in Caffe, with the additional parameter $B$ in SGHMC (Algorithm 1) set to $(1 - m)$, where $m$ is the *moment* variable defined in the SGD algorithm in Caffe.

**Cifar10-Quick net for CIFAR10** The Cifar10-Quick net consists of 3 convolutional layers, 3 max pool layers and 3 ReLU nonlinear layers, followed by 2 fully connected layers. The CIFAR-10 dataset consists of 60,000 color images of size 32×32 in 10 classes, with 50,000 for training and 10,000 for testing. Similar to LeNet, default parameter setting specified in Caffe is used.

In these models, the test function is defined as the cross entropy of the *softmax* outputs $\{\mathbf{o}_1, \cdots, \mathbf{o}_N\}$ for test data $\{(\mathbf{d}_1, y_1), \cdots, (\mathbf{d}_N, y_N)\}$ with $C$ classes, *i.e.*, loss $= -\sum_{i=1}^N \mathbf{o}_{y_i} + N \log \sum_{c=1}^C e^{\mathbf{o}_c}$. Since the theory indicates a linear speedup on the decrease of variance w.r.t. the number of workers, this means for a single run of the models, the loss would converge faster to its expectation with increasing number of workers. The following experiments verify this intuition.

### 5.2.1 Single-server experiments

We first test the single-server architecture in Section 4 on the three models. Because the expectations in the bias, MSE or variance are not analytically available in these complex models, we instead plot the *loss* versus *average number of iterations* ($\bar{L}$ defined in Section 4) on each worker and the running *time* in Figure 3. As mentioned above, faster decrease of the *loss* with more workers is expected.

For the ease of visualization, we only plot the results with $\{1, 2, 4, 6, 9\}$ workers; more detailed results are provided in Appendix I. We can see that generally the loss decreases faster with increasing number of workers. In the CIFAR-10 dataset, the final losses of 6 and 9 workers are worst than the one with 4 workers. It shows that the accuracy of the sample average suffers from the increased staleness due to the increased number of workers. Therefore a smaller step size $h$ should be considered to maintain high accuracy when using a large number of workers. Note the 1-worker curves correspond to the standard SG-MCMC, whose loss decreases much slower due to high estimation variance,

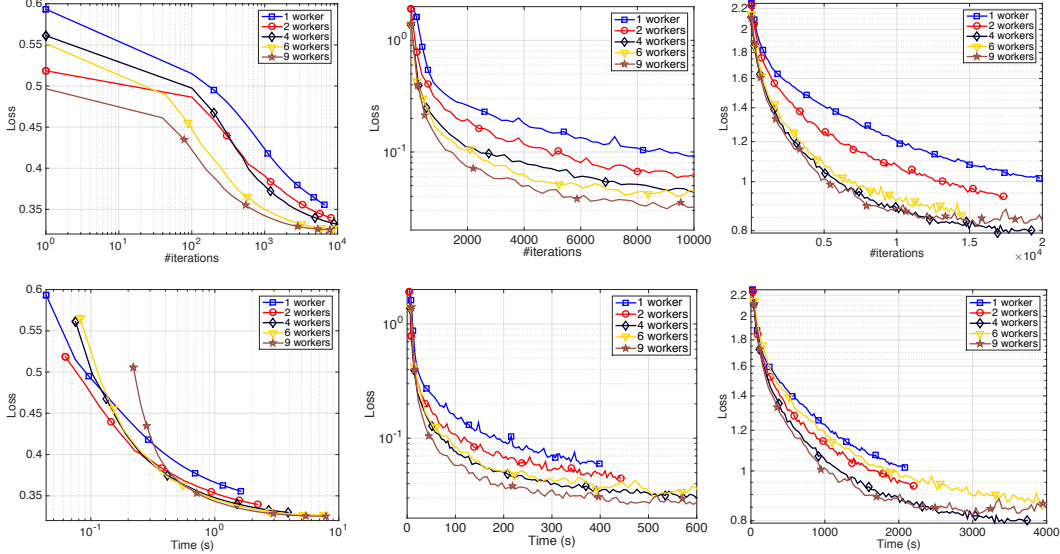

Figure 3: Testing loss vs. #workers. From left to right, each column corresponds to the a9a, MNIST and CIFAR dataset, respectively. The loss is defined in the text.

though in theory it has the same level of bias as the single-server architecture for a given number of iterations (they will converge to the same accuracy).

### 5.2.2 Multiple-server experiments

Finally, we test the multiple-servers architecture on the same models. We use the same criterion as the single-server setting to measure the convergence behavior. The *loss* versus *average number of iterations* on each worker ($\bar{L}$ defined in Section 4) for the three datasets are plotted in Figure 4, where we vary the number of servers among $\{1, 3, 5, 7\}$, and use 2 workers for each server. The plots of *loss* versus *time* and using different number of workers for each server are provided in the Appendix. We can see that in the simple BLR model, multiple servers do not seem to show significant speedup, probably due to the simplicity of the posterior, where the sample variance is too small for multiple servers to take effect; while in the more complicated deep neural networks, using more servers results in a faster decrease of the *loss*, especially in the MNIST dataset.

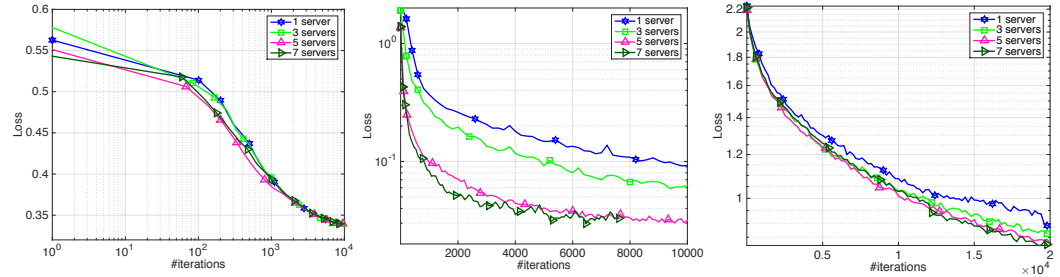

Figure 4: Testing loss vs. #servers. From left to right, each column corresponds to the a9a, MNIST and CIFAR dataset, respectively. The loss is defined in the text.

## 6 Conclusion

We extend theory from standard SG-MCMC to the stale stochastic gradient setting, and analyze the impacts of the staleness to the convergence behavior of an $S^2$G-MCMC algorithm. Our theory reveals that the estimation variance is independent of the staleness, leading to a linear speedup w.r.t. the number of workers, although in practice little speedup in terms of optimal bias and MSE might be achieved due to their dependence on the staleness. We test our theory on a simple asynchronous distributed SG-MCMC system with two simulated examples and several deep neural network models. Experimental results verify the effectiveness and scalability of the proposed $S^2$G-MCMC framework.

**Acknowledgements**   Supported in part by ARO, DARPA, DOE, NGA, ONR and NSF.

## Footnotes

*The existence of a nice $\psi$ is guaranteed in the elliptic/hypoelliptic SDE settings when $\mathbf{x}$ is on a torus [25].

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
