[Supplementary Material]

# Supplementary Material for:
# Stochastic Gradient MCMC with Stale Gradients

Changyou Chen[†]    Nan Ding[‡]    Chunyuan Li[†]    Yizhe Zhang[†]    Lawrence Carin[†]

[†]Dept. of Electrical and Computer Engineering, Duke University, Durham, NC, USA

[‡]Google Inc., Venice, CA, USA

[†]{cc448,cl319,yz196,lcarin}@duke.edu; [‡]dingnan@google.com

## A  A simple Bayesian distributed system based on S²G-MCMC

We provide the detailed architecture of the simple Bayesian distributed system described in Section 4. We put the single-chain and multiple-chain distributed SG-MCMCs into a unified framework. Suppose there are $S$ servers and $W$ workers, the one with $S = 1$ corresponds to the single-chain distributed SG-MCMC, whereas the one with $S > 1$ corresponds to the multiple-chain distributed SG-MCMC. The servers and workers are responsible for the following tasks:

- Each worker runs independently and communicates with a specific server. They are responsible for computing the stochastic gradients[§] of the parameter given by the server. Once the stochastic gradient is computed, the worker sends it to its assigned server and receive a new parameter sample from the server.

- Each server independently maintains its own state vector and timestamp. At the $l$-th timestamp[¶], it receives a stale stochastic gradient $\nabla_{\boldsymbol{\theta}} \hat{U}_{\tau_l}(\boldsymbol{\theta}) \triangleq \nabla_{\boldsymbol{\theta}} \tilde{U}(\boldsymbol{\theta}_{(l-\tau_l)h})$ from worker $w$, updates the state vector $\mathbf{x}_{lh}$ to $\mathbf{x}_{(l+1)h}$ and increments the timestamp, then sends the new parameter sample $\boldsymbol{\theta}_{(l+1)h}$ to worker $w$.

The sending and receiving in the servers and workers are performed asynchronously, enabling minimum communication cost and latency between the servers and workers. At testing, all the samples from the servers are collected and applied to a test function. Apparently, the training time using multiple servers is basically the same as using a single server because the sampling in different servers is independent. Figure 5 depicts the architecture of the proposed Bayesian distributed framework. Algorithm 2 details the algorithm on the servers and workers.

---

**Algorithm 2** Asynchronous Distributed SG-MCMC

---

**Server**

**Output:** $\{\mathbf{x}_h, \ldots, \mathbf{x}_{Lh}\}$

Initialize $\mathbf{x}_0 \in \mathbf{R}^m$;

Send $\boldsymbol{\theta}_0$ to all assigned workers;

**for** $l = 0, 1, \ldots, L - 1$ **do**

    Receive a stale stochastic gradient $\nabla \tilde{U}_{(l-\tau_l)h}$ from a worker $w$.

    Update $\mathbf{x}_{lh}$ to $\mathbf{x}_{(l+1)h}$ using $\nabla \tilde{U}_{(l-\tau_l)h}$. (*)

    Send $\boldsymbol{\theta}_{(l+1)h}$ to the worker $w$.

**end for**

---

**Worker**

**repeat**

    Receive $\boldsymbol{\theta}_{lh}$ from server $s$.

    Compute $\nabla \tilde{U}_{lh}$ with a minibatch.

    Send $\nabla \tilde{U}_{lh}$ to server $s$.

**until** $stop$

---

[§]This is the most expensive part in an SG-MCMC algorithm.

[¶]Each server is equipped with a timestamp because they are independent with each other.

Figure 5: Architecture of the proposed Bayesian distributed framework. In the multi-server case, the dash lines on the servers indicate a simple averaging operation for testing, otherwise the servers are independent. Section 3.3 provides more details.

The update rule (*) of the state vector in Algorithm 2 depends on which SG-MCMC algorithm is employed. For instance, Algorithm 1 describes the update rule of the SGHMC with a 1st-order Euler integrator.

## B   Assumptions

First, following [25], we will need to assume the corresponding SDE of SG-MCMC to be either elliptic or hypoelliptic. The ellipticity/hypoellipticity describes whether the Brownian motion is able to spread over the whole parameter space. The SDE of the SGLD is elliptic, while for other SG-MCMC algorithms such as the SGHMC, the hypoellipticity assumption is usually reasonable. When the domain $\mathbf{x}$ is on the torus, the ellipticity and hypoellipticity of an SDE guarantees the existence of a nice solution for the Poisson equation (5). The assumption is summarized in Assumption 2.

**Assumption 2.** *The corresponding SDE of a SG-MCMC algorithm is either elliptic or hypoelliptic$^{\parallel}$.*

When $\mathbf{x}$ is extended to the domain of $\mathbb{R}^p$ for some integer $p > 0$, we need some assumptions on the solution of the Poisson equation (5). Note (5) can be equivalently written in an integration form [35] using Itô's formula:

$$
\frac{1}{t} \int_0^t \phi(\mathbf{x}_s) \mathrm{d}s - \bar{\phi} \tag{6}
$$
$$
= \frac{1}{t} \left( \psi(\mathbf{x}_t) - \psi(\mathbf{x}_0) \right) - \frac{1}{t} \int_0^t \nabla \psi(\mathbf{x}_s) \cdot g(\mathbf{x}_s) \mathrm{d}\mathbf{w}_s \ .
$$

Intuitively, $\psi$ needs to be bounded if the discrepancy between $\hat{\phi}_L$ and $\bar{\phi}$ were to be bounded. This is satisfied if the SDE is defined in a bounded domain [25]. In the unbounded domain as for SG-MCMC algorithms, it turns out the following boundedness assumptions on $\psi$ suffice [17].

**Assumption 3.** *1) $\psi$ and its up to 3rd-order derivatives, $\mathcal{D}^k\psi$, are bounded by a function $\mathcal{V}$, i.e., $\|\mathcal{D}^k\psi\| \leq C_k \mathcal{V}^{p_k}$ for $k = (0, 1, 2, 3)$, $C_k, p_k > 0$. 2) the expectation of $\mathcal{V}$ on $\{\mathbf{x}_{lh}\}$ is bounded: $\sup_l \mathbb{E}\mathcal{V}^p(\mathbf{x}_{lh}) < \infty$. 3) $\mathcal{V}$ is smooth such that $\sup_{s \in (0,1)} \mathcal{V}^p \left( s\,\mathbf{x} + (1-s)\,\mathbf{y} \right) \leq C \left( \mathcal{V}^p \left( \mathbf{x} \right) + \mathcal{V}^p \left( \mathbf{y} \right) \right), \forall \mathbf{x}, \mathbf{y}, p \leq \max\{2p_k\}$ for some $C > 0$.*

Furthermore, in our proofs the expectation of a function under a diffusion needs to be expanded in a Taylor expansion style, *e.g.*, $\mathbb{E}\phi(\mathbf{x}_t) = \sum_{i=0}^{\ell} \frac{t^i}{i!} \mathcal{L}^i \phi(\mathbf{x}_0) + t^{\ell+1} r_{\ell,F,\phi}(\mathbf{x}_0)$ by using Kolmogorov's

---

$^{\parallel}$The SDE of the SGLD can be verified to be elliptic. For other SG-MCMC algorithms such as the SGHMC, the hypoellipticity assumption is usually reasonable, see [25] on how to verify hypoellipticity of an SDE.

backward equation. To ensure the remainder term $r_{\ell,F,\phi}(\mathbf{x}_0)$ to be bounded, it suffices to make the following assumption on the smoothness and boundedness of $F(\mathbf{x})$ [35, 17].

**Assumption 4.** $F(\mathbf{x})$ *is infinitely differentiable with bounded derivatives of any order; and* $|F(\mathbf{x})| \leq A(1 + |\mathbf{x}|^s)$ *for some integer* $s > 0$ *and* $A > 0$.

## C   Notation

For simplicity, we will simplify some notation used in the proof as follows:

$$\nabla_{\boldsymbol{\theta}} \tilde{U}_l(\boldsymbol{\theta}_{lh}) \triangleq \nabla_{\boldsymbol{\theta}} \tilde{U}_{lh} \triangleq \tilde{G}_{lh}$$

$$\nabla_{\boldsymbol{\theta}} U_l(\boldsymbol{\theta}_{lh}) \triangleq \nabla_{\boldsymbol{\theta}} U_{lh} \triangleq G_{lh}$$

$$\psi(\mathbf{X}_{lh}) \triangleq \psi_{lh}$$

## D   Proof of Theorem 2

In S$^2$G-MCMC, for the $l$-th iteration, suppose a stochastic gradient with a staleness $\tau_l$ is used, *e.g.*, $\tilde{G}_{(l-\tau_l)h}$. First, we will bound the difference between $\tilde{G}_{(l-\tau_l)h}$ and the stochastic gradient at the $l$-th iteration $\tilde{G}_{lh}$, by using the Lipschitz property of $\tilde{G}_{lh}$, with the following lemma.

**Lemma 8.** *Let* $f_{lh} \triangleq \left\| \mathbf{x}_{lh} - \mathbf{x}_{(l-1)h} \right\|$, *the expected difference between* $\tilde{G}_{(l-\tau_l)h}$ *and* $\tilde{G}_{lh}$ *is bounded by:*

$$\left\| \mathbb{E}\left( \tilde{G}_{(l-\tau_l)h} - \tilde{G}_{lh} \right) \right\| = \max_{i=l-\tau_l}^{l-1} |\mathcal{L}_i f_{ih}| C\tau h + O(h^2), \tag{7}$$

*where the expectation is taken over the randomness of the SG-MCMC algorithm, e.g., the randomness from stochastic gradients and the injected Gaussian noise.*

*Proof.* Note the randomness of $\tilde{G}_{lh}$ comes from two sources, the injected Gaussian noise and the stochastic gradient noise. We denote the expectations with respect to these two randomness as $\mathbb{E}_\zeta$ and $\mathbb{E}_g$, respectively. The whole expectation thus can be decomposed as $\mathbb{E} = \mathbb{E}_\zeta \mathbb{E}_g$.

Applying the Lipschitz property of $\tilde{G}_{lh}$, we have

$$\left\| \mathbb{E}\left( \tilde{G}_{(l-\tau_l)h} - \tilde{G}_{lh} \right) \right\| = \left\| \mathbb{E}_\zeta \left( G_{(l-\tau_l)h} - G_{lh} \right) \right\|$$

$$\leq \mathbb{E}_\zeta \left\| \left( G_{(l-\tau_l)h} - G_{lh} \right) \right\|$$

$$\leq C \mathbb{E}_\zeta \left\| \left( \boldsymbol{\theta}_{(l-\tau_l)h} - \theta_{lh} \right) \right\|$$

$$\leq C \mathbb{E}_\zeta \left\| \sum_{i=l-\tau_l}^{l-1} \left( \boldsymbol{\theta}_{(ih)} - \boldsymbol{\theta}_{(i+1)h} \right) \right\|$$

$$\leq C \sum_{i=l-\tau_l}^{l-1} \mathbb{E}_\zeta \left\| \left( \boldsymbol{\theta}_{(ih)} - \boldsymbol{\theta}_{(i+1)h} \right) \right\|$$

$$\leq C \sum_{i=l-\tau_l}^{l-1} \mathbb{E}_\zeta \left\| \mathbf{x}_{(i+1)h} - \mathbf{x}_{ih} \right\|$$

From the definition of $K$th-order integrator, *i.e.*, $\mathbb{E}_\zeta f(\mathbf{x}_{lh}) = e^{\tilde{\mathcal{L}}_l h} f(\mathbf{x}_{(l-1)h}) + O(h^{K+1})$, if we let

$$f(\mathbf{x}_{lh}) = \left\| \mathbf{x}_{lh} - \mathbf{x}_{(l-1)h} \right\| \triangleq f_{lh},$$

where $\mathbf{x}_{(l-1)h}$ is the starting point in the $l$-th iteration, and note that

$$f(\mathbf{x}_{(l-1)h}) = 0.$$

We have

$$C \sum_{i=l-\tau_l}^{l-1} \mathbb{E}_\zeta \left\| \mathbf{x}_{(i+1)h} - \mathbf{x}_{ih} \right\| \triangleq C \sum_{i=l-\tau_l}^{l-1} \mathbb{E}_\zeta f(\mathbf{x}_{lh}) \tag{8}$$

$$\leq C \sum_{i=l-\tau_l}^{l-1} \left( e^{\mathcal{L}_i h} f(\mathbf{x}_{(i-1)h}) + O(h^{K+1}) \right) \tag{9}$$

$$\leq C \sum_{i=l-\tau_l}^{l-1} |\mathcal{L}_i f_{ih}| \, h + O(h^2) \tag{10}$$

$$\leq \max_{i=l-\tau_l}^{l-1} |\mathcal{L}_i f_{ih}| \, C\tau h + O(h^2) \,,$$

where (10) is obtained by expanding the exponential operator and the assumption that the high order terms are bounded. □

Now we proceed to prove Theorem 2. The basic technique follows [17], thus we skip some derivations for some steps.

*Proof of Theorem 2.* Before the proof, let us first define some notation. First, define the operator $\Delta V_l$ for each $l$ as a differential operator as for any function $\psi$:

$$\Delta V_l \psi \triangleq \left( \tilde{G}_{l-\tau_l} - G_l \right) \cdot \nabla_{\mathbf{p}} \psi \,.$$

Second, define the local generator, $\tilde{\mathcal{L}}_l$, for an Itô diffusion, where the true gradient in (1) is replaced with the stochastic gradient from the $l$-th iteration, *i.e.*, $\tilde{\mathcal{L}}_l f(\mathbf{X}_t) \triangleq$

$$\left( \tilde{F}_l(\mathbf{x}_t) \cdot \nabla + \frac{1}{2} \left( \sigma(\mathbf{x}_t)\sigma(\mathbf{x}_t)^T \right) : \nabla \nabla^T \right) f(\mathbf{x}_t) \,,$$

for a compactly supported twice differentiable function $f$, where $\tilde{F}_l$ is the same as $F$ but with the full gradient $\tilde{G}_{lh}$ replaced with the stochastic gradient $\tilde{G}_{lh}$. Based on these definitions, we have

$$\tilde{\mathcal{L}}_l = \mathcal{L} + \Delta V_l \,.$$

Following [17], for an SG-MCMC with a $K$th-order integrator, and a test function $\phi$, we have:

$$\mathbb{E}[\psi(\mathbf{x}_{lh})] = \left( \mathbb{I} + h\tilde{\mathcal{L}}_l \right) \psi(\mathbf{x}_{(l-1)h}) \tag{11}$$

$$+ \sum_{k=2}^{K} \frac{h^k}{k!} \tilde{\mathcal{L}}_l^k \psi(\mathbf{x}_{(l-1)h}) + O\left( \frac{h^{K+1}}{(K+1)!} \tilde{\mathcal{L}}_l^{K+1} \psi_{(l-1)h} \right) \,,$$

where $\mathbb{I}$ is the identity map. Sum over $l = 1, \cdots, L$ in (11), take expectation on both sides, and use the relation $\tilde{\mathcal{L}}_l = \mathcal{L} + \Delta V_l$ to expand the first order term. We obtain

$$\sum_{l=1}^{L} \mathbb{E}[\psi(\mathbf{x}_{lh})] = \psi(\mathbf{x}_0) + \sum_{l=1}^{L-1} \mathbb{E}[\psi(\mathbf{x}_{lh})]$$

$$+ h \sum_{l=1}^{L} \mathbb{E}[\mathcal{L}\psi(\mathbf{x}_{(l-1)h})] + h \sum_{l=1}^{L} \mathbb{E}[\Delta V_l \psi(\mathbf{x}_{(l-1)h})]$$

$$+ \sum_{k=2}^{K} \frac{h^k}{k!} \sum_{l=1}^{L} \mathbb{E}[\tilde{\mathcal{L}}_l^k \psi(\mathbf{x}_{(l-1)h})]$$

$$+ O\left( \frac{h^{K+1}}{(K+1)!} \sum_{l} \mathbb{E}\tilde{\mathcal{L}}_l^{K+1} \psi_{(l-1)h} \right) \,.$$

Divide both sides by $Lh$, use the Poisson equation (5), and reorganize terms. We have:

$$\mathbb{E}[\frac{1}{L}\sum_l \phi(\mathbf{x}_{lh}) - \bar{\phi}] = \frac{1}{L}\sum_{l=1}^{L}\mathbb{E}[\mathcal{L}\psi(\mathbf{x}_{(l-1)h})] \tag{12}$$

$$=\frac{1}{Lh}\left(\mathbb{E}[\psi(\mathbf{x}_{lh})] - \psi(\mathbf{x}_0)\right) - \frac{1}{L}\sum_l \mathbb{E}[\Delta V_l \psi(\mathbf{x}_{(l-1)h})]$$

$$-\sum_{k=2}^{K}\frac{h^{k-1}}{k!L}\sum_{l=1}^{L}\mathbb{E}[\tilde{\mathcal{L}}_l^k \psi(\mathbf{x}_{(l-1)h})] + O\left(\frac{h^K}{(K+1)!L}\sum_l \mathbb{E}\tilde{\mathcal{L}}_l^{K+1}\psi_{(l-1)h}\right)$$

According to [17], the term $\sum_l \mathbb{E}[\tilde{\mathcal{L}}_l^k \psi(\mathbf{x}_{(l-1)h})]$ is bounded by $\sum_l \mathbb{E}[\tilde{\mathcal{L}}_l^k \psi(\mathbf{X}_{(l-1)h})]$

$$= O\left(\frac{1}{h} + h^{K-k+1}\sum_l \mathbb{E}\tilde{\mathcal{L}}_l^{K+1}\psi_{(l-1)h}\right), \tag{13}$$

Substituting (13) into (12), after simplification, we have: $\mathbb{E}\left(\frac{1}{L}\sum_l \phi(\mathbf{x}_{lh}) - \bar{\phi}\right)$

$$=\frac{1}{Lh}\underbrace{\left(\mathbb{E}[\psi(\mathbf{x}_{lh})] - \psi(\mathbf{x}_0)\right)}_{C_1} - \frac{1}{L}\underbrace{\sum_l \mathbb{E}[\Delta V_l \psi(\mathbf{x}_{(l-1)h})]}_{C_2}$$

$$-\sum_{k=2}^{K}O\left(\frac{h^{k-1}}{Lh} + \frac{h^K}{L}\sum_l \frac{1}{k!}\mathbb{E}\tilde{\mathcal{L}}_l^K \psi_{(l-1)h}\right) + \frac{h^K}{(K+1)!L}\sum_l \mathbb{E}\tilde{\mathcal{L}}_l^{K+1}\psi_{(l-1)h},$$

According to the assumption, the term $C_1$ is bounded. For term $C_2$, according to the Cauchy–Schwarz inequality, we have

$$|C_2| = \frac{1}{L}\left|\sum_l \mathbb{E}\left(\tilde{G}_{(l-\tau_l)h} - G_{lh}\right)\cdot\mathbb{E}\nabla\psi_{(l-1)h}\right|$$

$$\leq\frac{1}{L}\sum_l\left|\mathbb{E}\left(\tilde{G}_{(l-\tau_l)h} - G_{lh}\right)\cdot\mathbb{E}\nabla\psi_{(l-1)h}\right|$$

$$\leq\frac{1}{L}\sum_l\left\|\mathbb{E}\left(\tilde{G}_{(l-\tau_l)h} - G_{lh}\right)\right\|\left\|\mathbb{E}\nabla\psi_{(l-1)h}\right\|$$

$$\leq\frac{1}{L}\sum_l\left(\left\|\mathbb{E}\left(\tilde{G}_{(l-\tau_l)h} - \tilde{G}_{lh}\right)\right\| + \left\|\mathbb{E}\left(\tilde{G}_{lh} - G_{lh}\right)\right\|\right)\left\|\mathbb{E}\nabla\psi_{(l-1)h}\right\|$$

$$=\frac{1}{L}\sum_l\left\|\mathbb{E}\left(\tilde{G}_{(l-\tau_l)h} - \tilde{G}_{lh}\right)\right\|\left\|\mathbb{E}\nabla\psi_{(l-1)h}\right\|$$

Applying (7) from Lemma 8, we have

$$|C_2| \leq \frac{1}{L}\sum_l\left(\max_{i=l-\tau_l}^{l}\|\mathcal{L}_i\|\left\|\mathbb{E}\nabla\psi_{lh}\right\| C\tau_l h\right)$$

$$\leq \max_l\|\mathcal{L}_l\|\max_l\left\|\mathbb{E}\nabla\psi_{lh}\right\| C\tau h .$$

As a result, collecting low order terms, the bias can be expressed as:

$$\left|\mathbb{E}\hat{\phi} - \bar{\phi}\right| = \left|\mathbb{E}\left(\frac{1}{L}\sum_l \phi(\mathbf{x}_{lh}) - \bar{\phi}\right)\right|$$

$$=\left|\frac{C_1}{Lh} - C_2 + h^K\sum_{k=1}^{K}\frac{1}{(k+1)!L}\sum_l \mathbb{E}\tilde{\mathcal{L}}_l^{k+1}\psi_{(l-1)h}\right| .$$

$$\tag{14}$$

As a result, there exists some constant $D_1$ independent of $(L, h, \tau)$, such that

$$\left|\mathbb{E}\hat{\phi} - \bar{\phi}\right| \leq D_1 \left|\frac{1}{Lh}\right| + |C_2| + \left|M_1\tau h + \left|M_2 h^K\right|\right| \tag{15}$$

$$= D_1\left(\frac{1}{Lh} + M_1\tau h + M_2 h^K\right),$$

where $M_1 \triangleq \max_l \|\mathcal{L}_l\| \max_l \|\mathbb{E}\nabla\psi_{lh}\| C$, $M_2 \triangleq \sum_{k=1}^K \frac{1}{(k+1)!L} \sum_l \mathbb{E}\tilde{\mathcal{L}}_l^{k+1}\psi_{(l-1)h}$. (15) follows by substituting the inequality for $C_2$ above. This completes the proof. $\qquad\square$

## E  Proof of Theorem 3

*Proof.* Similar to the proof of Theorem 2, we first expand $\mathbb{E}\psi_{lh}$ using the property of $K$th-order integrator as

$$\sum_{l=1}^L \mathbb{E}\left(\psi(\mathbf{x}_{lh})\right) = \sum_{l=1}^L \psi(\mathbf{x}_{(l-1)h}) + h\sum_{l=1}^L \mathcal{L}\psi(\mathbf{x}_{(l-1)h})$$

$$+ h\sum_{l=1}^L \Delta V_l \psi(\mathbf{x}_{(l-1)h}) + \sum_{k=2}^K \frac{h^k}{k!}\sum_{l=1}^L \tilde{\mathcal{L}}_l^k \psi(\mathbf{x}_{(l-1)h})$$

$$+ O\left(\frac{h^{K+1}}{(K+1)!}\sum_l \tilde{\mathcal{L}}_l^{K+1}\psi_{(l-1)h}\right).$$

Substituting the Poisson equation (5) into the above equation, dividing both sides by $Lh$ and rearranging related terms arrives

$$\hat{\phi} - \bar{\phi} = \frac{1}{Lh}\left(\mathbb{E}\psi(\mathbf{x}_{Lh}) - \psi(\mathbf{x}_0)\right) \tag{16}$$

$$- \frac{1}{Lh}\sum_{l=1}^L \left(\mathbb{E}\psi_{(l-1)h} - \psi_{(l-1)h}\right) - \frac{1}{L}\sum_{l=1}^L \Delta V_l \psi_{(l-1)h}$$

$$- \sum_{k=2}^K \frac{h^{k-1}}{2L}\sum_{l=1}^L \tilde{\mathcal{L}}_l^k \psi(\mathbf{x}_{(l-1)h}) + O\left(\frac{h^K}{L(K+1)!}\sum_l \tilde{\mathcal{L}}_l^{K+1}\psi_{(l-1)h}\right)$$

Taking square on both sides, we have there exists some positive constant $D$, such that

$$\left(\hat{\phi} - \bar{\phi}\right)^2 \leq D\left(\underbrace{\frac{(\mathbb{E}\psi_{Lh} - \psi_0)^2}{L^2 h^2}}_{A_1} + \underbrace{\frac{1}{L^2 h^2}\sum_{l=1}^L \left(\mathbb{E}\psi_{(l-1)h} - \psi_{(l-1)h}\right)^2}_{A_2}\right.$$

$$\left. + \underbrace{\left(\frac{1}{L}\sum_{l=1}^L \Delta V_l \psi_{(l-1)h}\right)^2}_{A_3} + \underbrace{\sum_{k=2}^K \frac{h^{2(k-1)}}{k!L^2}\left(\sum_{l=1}^L \tilde{\mathcal{L}}_l^k \psi_{(l-1)h}\right)^2}_{A_4} + \underbrace{\left(\frac{\sum_l \tilde{\mathcal{L}}_l^{K+1}\psi_{(l-1)h}}{L(K+1)!}\right)^2 h^{2K}}_{A_5}\right) \tag{17}$$

After taking expectation, we have

$$\mathbb{E}\left(\hat{\phi} - \bar{\phi}\right)^2 \leq C\left(\mathbb{E}A_1 + \mathbb{E}A_2 + \mathbb{E}A_3 + \mathbb{E}A_4 + \mathbb{E}A_5\right)$$

$A_1$ is easily bounded by the assumption that $\|\psi\| \leq V^{p_0} < \infty$. From the proof of Theorem 3 in [17], $A_2$ and $A_4$ are also bounded, which are summarized in Lemma 9.

**Lemma 9.** *The terms $\mathbb{E}A_2$ and $\mathbb{E}A_4$ are bounded by:*

$$\mathbb{E}A_2 = O\left(\frac{1}{Lh}\right)$$

$$\mathbb{E}A_4 = O\left(\frac{1}{Lh} + h^{2K}\sum_{k=2}^{K}\frac{1}{Lk!}\sum_l \tilde{\mathcal{L}}_l^{k+1}\psi_{(l-1)h}\right) \ .$$

We are left to show a bound for $\mathbb{E}A_3$. First we have

$$\mathbb{E}A_3 = \mathbb{E}\left(\frac{1}{L}\sum_{l=1}^{L}\Delta V_l \psi_{(l-1)h}\right)^2$$

$$=\mathbb{E}\left(\frac{1}{L}\sum_{l=1}^{L}\left(\tilde{G}_{(l-\tau_l)h} - G_{lh}\right)\cdot\nabla_{\mathbf{p}}\psi_{(l-1)h}\right)^2$$

$$=\frac{1}{L^2}\sum_{i=1}^{L}\sum_{j=1}^{L}\mathbb{E}\left[\left(\tilde{G}_{(i-\tau_i)h} - G_{ih}\right)\cdot\nabla_{\mathbf{p}}\psi_{(i-1)h}\left(\tilde{G}_{(j-\tau_j)h} - G_{jh}\right)\cdot\nabla_{\mathbf{p}}\psi_{(j-1)h}\right]$$

Using the Cauchy–Schwartz inequality, we have

$$\leq\frac{1}{L^2}\sum_{i=1}^{L}\sum_{j=1}^{L}\left\|\mathbb{E}\left(\tilde{G}_{(i-\tau_i)h} - G_{ih}\right)\right\|\left\|\mathbb{E}\left(\tilde{G}_{(j-\tau_j)h} - G_{jh}\right)\right\|\left\|\mathbb{E}\nabla\psi_{(i-1)h}\right\|\left\|\mathbb{E}\nabla\psi_{(j-1)h}\right\|$$

$$\leq\frac{1}{L^2}\sum_{i=1}^{L}\sum_{j=1}^{L}\left(\left\|\mathbb{E}\left(\tilde{G}_{(i-\tau_i)h} - \tilde{G}_{ih}\right)\right\| + \left\|\mathbb{E}\left(\tilde{G}_{ih} - G_{ih}\right)\right\|\right)$$

$$\left(\left\|\mathbb{E}\left(\tilde{G}_{(j-\tau_j)h} - \tilde{G}_{jh}\right)\right\| + \left\|\mathbb{E}\left(\tilde{G}_{jh} - G_{jh}\right)\right\|\right)\left\|\mathbb{E}\nabla\psi_{(i-1)h}\right\|\left\|\mathbb{E}\nabla\psi_{(j-1)h}\right\|$$

$$=\frac{1}{L^2}\sum_{i=1}^{L}\sum_{j=1}^{L}\left\|\mathbb{E}\left(\tilde{G}_{(i-\tau_i)h} - \tilde{G}_{ih}\right)\right\|\left\|\mathbb{E}\left(\tilde{G}_{(j-\tau_j)h} - \tilde{G}_{jh}\right)\right\|\left\|\mathbb{E}\nabla\psi_{(i-1)h}\right\|\left\|\mathbb{E}\nabla\psi_{(j-1)h}\right\|$$

Applying (7) from Lemma 8, we have

$$\mathbb{E}A_3 \leq \max_l\left\|\mathbb{E}\nabla\psi_{lh}\right\|^2\max_l\left(\mathcal{L}_l f_{lh}\right)^2 C^2\tau^2 h^2 \ .$$

Collecting low order terms from the above bounds, we have there exists some constant $D_2$ independent of $(L, h, \tau)$, such that

$$\mathbb{E}\left(\hat{\phi} - \bar{\phi}\right)^2$$

$$\leq\frac{C_1}{Lh} + C_2 h^{2K} + \max_l\left\|\mathbb{E}\nabla\psi_{lh}\right\|^2\max_l\left\|\mathcal{L}_l\right\|^2 C^2\tau^2 h^2$$

$$\leq D_2\left(\frac{1}{Lh} + \tilde{M}_1\tau^2 h^2 + \tilde{M}_2 h^{2K}\right) \ ,$$

where $\tilde{M}_1 \triangleq \max_l\left\|\mathbb{E}\nabla\psi_{lh}\right\|^2\max_l\left(\mathcal{L}_l f_{lh}\right)^2 C^2$, $\tilde{M}_2 \triangleq \mathbb{E}\left(\frac{1}{L(K+1)!}\sum_l\tilde{\mathcal{L}}_l^{K+1}\psi_{(l-1)h}\right)^2$. This completes the proof. □

## F Proof of Theorem 4

In the proof, we will use the following simple result stated Lemma 10.

**Lemma 10.** *Let $(\mathcal{M}_1, \cdots, \mathcal{M}_N)$ be a set of independent martingale, i.e, $\mathbb{E}\left[\mathcal{M}_n|\mathcal{F}\right] = 0$, where $\mathcal{F}$ is the filtration generated by $\mathcal{M}_n$. Then we have*

$$\mathbb{E}\left[\left(\sum_{n=1}^{N}\mathcal{M}_n\right)^2 \bigg| \mathcal{F}\right] = \sum_{n=1}^{N}\mathbb{E}\left[\mathcal{M}_n^2|\mathcal{F}\right] \ . \tag{18}$$

*Proof.*

$$\mathbb{E}\left[\left(\sum_{n=1}^{N}\mathcal{M}_n\right)^2|\mathcal{F}\right] = \mathbb{E}\left[\sum_{i=1}^{N}\sum_{j=1}^{N}\mathcal{M}_i\mathcal{M}_j|\mathcal{F}\right]$$

$$=\mathbb{E}\left[\sum_{i=1}^{N}\mathcal{M}_i^2|\mathcal{F}\right] + \sum_{i\neq j}\mathbb{E}\left[\mathcal{M}_i|\mathcal{F}\right]\mathbb{E}\left[\mathcal{M}_j|\mathcal{F}\right]$$

$$=\sum_{i=1}^{N}\mathbb{E}\left[\mathcal{M}_i^2|\mathcal{F}\right] .$$

$\square$

In the following we will omitted the filtration $\mathcal{F}$ in the expectation for simplicity. We we now ready to prove Theorem 4.

*Proof.* By definition, we have

$$\mathrm{Var}\left(\hat{\phi}_L\right) = \mathbb{E}\left(\hat{\phi}_L - \bar{\phi} - \left(\mathbb{E}\hat{\phi}_L - \bar{\phi}\right)\right)^2$$

Substitute (12) and (16) into the above equation, we have

$$\hat{\phi}_L - \mathbb{E}\bar{\phi} = -\frac{1}{Lh}\sum_l\left(\mathbb{E}\psi_{(l-1)h} - \psi_{(l-1)h}\right)$$

$$-\frac{1}{L}\sum_l\left(A_1 - \mathbb{E}A_1\right) - \sum_k\frac{h^{k-1}}{k!L}\sum_l\left(A_2 - \mathbb{E}A_2\right) - \frac{h^K}{(K+1)!L}\sum_l\left(A_3 - \mathbb{E}A_3\right) ,$$

where

$$A_1 \triangleq \Delta V_l\psi_{(l-1)h}$$
$$A_2 \triangleq \tilde{\mathcal{L}}_l^k\psi_{(l-1)h}$$
$$A_3 \triangleq \tilde{\mathcal{L}}_l^{K+1}\psi_{(l-1)h} .$$

Take square on both sides, following by expectation, and note that all $(A_i - \mathbb{E}A_i)$ are martingale for $i = 1, 2, 3$, which allows us to use (18) from Lemma 10. We have there exists a constant $D$ independent of $(L, h, \tau)$, such that

$$\mathrm{Var}\left(\hat{\phi}_L\right) \leq D\left(\frac{1}{L^2h^2}\mathbb{E}\left(\sum_l\left(\mathbb{E}\psi_{(l-1)h} - \psi_{(l-1)h}\right)\right)^2\right.$$

$$+\frac{1}{L^2}\sum_l\mathbb{E}\left(A_1 - \mathbb{E}A_1\right)^2 + \sum_k\frac{h^{2(k-1)}}{(k!L)2}\sum_l\mathbb{E}\left(A_2 - \mathbb{E}A_2\right)^2$$

$$\left.+\frac{h^{2K}}{((K+1)!L)^2}\sum_l\mathbb{E}\left(A_3 - \mathbb{E}A_3\right)^2\right)$$

$$\leq D\left(\underbrace{\frac{1}{L^2h^2}\mathbb{E}\left(\sum_l\left(\mathbb{E}\psi_{(l-1)h} - \psi_{(l-1)h}\right)\right)^2}_{B_1}\right.$$

$$\left.+\frac{1}{L^2}\sum_l\mathbb{E}\left(A_1 - \mathbb{E}A_1\right)^2 + \sum_{k=2}^{K}\frac{h^{2(k-1)}}{(k!L)2}\sum_l\mathbb{E}A_2^2 + \frac{h^{2K}}{((K+1)!L)^2}\sum_l\mathbb{E}A_3^2\right) .$$

According to Lemma 9, $B_1$ is bounded by

$$B_1 = O\left(\frac{1}{Lh}\right).$$

Furthermore, according to the assumptions, both $\mathbb{E}A_2^2$ and $\mathbb{E}A_3^2$ are bounded. The delayed parameter $\tau$ exists in $\mathbb{E}\left(A_1 - \mathbb{E}A_1\right)^2$, we have

$$\begin{aligned}
&\mathbb{E}\left(A_1 - \mathbb{E}A_1\right)^2 \\
=&\mathbb{E}\left(\Delta V_l \psi_{(l-1)h} - \mathbb{E}\Delta V_l \psi_{(l-1)h}\right)^2 \\
=&\mathbb{E}\left(\left(\tilde{G}_{(l-\tau_l)h} - G_{lh}\right) \cdot \nabla_{\mathbf{p}}\psi_{(l-1)h} - \mathbb{E}\left(\tilde{G}_{(l-\tau_l)h} - G_{lh}\right) \cdot \nabla_{\mathbf{p}}\psi_{(l-1)h}\right)^2
\end{aligned}$$

Expanding the terms, we have there exists a constant $D_1$ such that

$$\begin{aligned}
&\mathbb{E}\left(A_1 - \mathbb{E}A_1\right)^2 \\
\leq& D_1 \mathbb{E}\left(\tilde{G}_{(l-\tau_l)h} \cdot \nabla_{\mathbf{p}}\psi_{(l-1)h} - \mathbb{E}\tilde{G}_{(l-\tau_l)h} \cdot \nabla_{\mathbf{p}}\psi_{(l-1)h}\right)^2 \\
&+ D_1 \mathbb{E}\left(G_{lh} \cdot \nabla_{\mathbf{p}}\psi_{(l-1)h} - \mathbb{E}G_{lh} \cdot \nabla_{\mathbf{p}}\psi_{(l-1)h}\right)^2 \\
=& D_1 \mathbb{E}\left(\tilde{G}_{(l-\tau_l)h} \cdot \nabla_{\mathbf{p}}\psi_{(l-1)h}\right)^2 + D_1 \mathbb{E}\left(G_{lh} \cdot \left(\nabla_{\mathbf{p}}\psi_{(l-1)h} - \nabla_{\mathbf{p}}\psi_{(l-1)h}\right)\right)^2 \\
\leq& D_1 \left(\mathbb{E}\left\|\tilde{G}_{(l-\tau_l)h}\right\|^2 \mathbb{E}\left\|\nabla_{\mathbf{p}}\psi_{(l-1)h}\right\|^2 + \mathbb{E}\left\|G_{lh}\right\|^2 \mathbb{E}\left\|\nabla_{\mathbf{p}}\psi_{(l-1)h} - \nabla_{\mathbf{p}}\psi_{(l-1)h}\right\|^2\right) \\
\leq& D_1 \sup_l \left\{\mathbb{E}\left\|\tilde{G}_{lh}\right\|^2 \mathbb{E}\left\|\nabla_{\mathbf{p}}\psi_{lh}\right\|^2 + \mathbb{E}\left\|G_{lh}\right\|^2 \mathbb{E}\left\|\nabla_{\mathbf{p}}\psi_{lh}\right\|^2\right\}.
\end{aligned}$$

According to the assumptions, the above bound is bounded, and does not depend on $\tau$. As a result,

$$\frac{1}{L^2}\sum_l \mathbb{E}\left(A_1 - \mathbb{E}A_1\right)^2 \leq \frac{D_1}{L}.$$

In addition, the bounds for both $\mathbb{E}A_2^2$ and $\mathbb{E}A_3^2$ are given in Lemma 9, which are higher-order terms with respect to $h$, *i.e.*, $O\left(h^{2K}\right)$.

Collecting low order terms, we have there exists a constant $D$ independent of $(L, h, \tau)$, such that the variance is bounded by:

$$\mathrm{Var}\left(\hat{\phi}_L\right) \leq D\left(\frac{1}{Lh} + h^{2K}\right) = D\left(\frac{1}{W\bar{L}h} + h^{2K}\right).$$

$\square$

# G  Proof of Theorem 6

We separate the proof for the bias and MSE, respectively.

*Proof for the bias.* According to the definition of $\hat{\phi}_L^S$, we have

$$\left| \mathbb{E}\hat{\phi}_L^S - \bar{\phi} \right| = \left| \mathbb{E} \sum_{s=1}^{S} \frac{T_s}{T} \hat{\phi}_{L_s} - \bar{\phi} \right|$$

$$= \left| \sum_{s=1}^{S} \frac{T_s}{T} \mathbb{E} \left( \hat{\phi}_{L_s} - \bar{\phi} \right) \right|$$

$$\leq \sum_{s=1}^{S} \frac{T_s}{T} \left| \mathbb{E}\hat{\phi}_{L_s} - \bar{\phi} \right|$$

$$= \sum_{s=1}^{S} \frac{T_s}{T} D_1 \left( \frac{1}{L_s h_s} + \left( M_1 \tau h_s + M_2 h_s^K \right) \right) \qquad (19)$$

$$= D_1 \left( \frac{S}{T} + \sum_{s=1}^{S} \frac{T_s}{T} \left( M_1 \tau h_s + M_2 h_s^K \right) \right)$$

$$\leq D_1 \left( \frac{S}{T} + \frac{S T_m}{T} \left( M_1 \tau h_m + M_2 h_m^K \right) \right) , \qquad (20)$$

where $T_m \triangleq \max_l T_l$, $h_m \triangleq \max_l h_l$, (19) follows by substituting the bias from Theorem 2 for each server into the formula. □

Similarly, for the MSE bound, we have

$$\mathbb{E} \left( \hat{\phi}_L^S - \bar{\phi} \right)^2 = \mathbb{E} \left( \sum_{s=1}^{S} \frac{T_s}{T} \left( \hat{\phi}_{L_s} - \bar{\phi} \right) \right)^2$$

$$= \sum_{s=1}^{S} \frac{T_s^2}{T^2} \mathbb{E} \left( \hat{\phi}_{L_s} - \bar{\phi} \right)^2 + \sum_{i \neq j} \frac{T_i T_j}{T_2} \mathbb{E} \left[ \hat{\phi}_{L_i} - \bar{\phi} \right] \mathbb{E} \left[ \hat{\phi}_{L_j} - \bar{\phi} \right]$$

$$\leq \sum_{s=1}^{S} \frac{T_s^2}{T^2} \mathbb{E} \left( \hat{\phi}_{L_s} - \bar{\phi} \right)^2 + \sum_{i \neq j} \frac{T_i T_j}{T^2} \left| \mathbb{E}\hat{\phi}_{L_i} - \bar{\phi} \right| \left| \mathbb{E}\hat{\phi}_{L_j} - \bar{\phi} \right| .$$

Substituting the bounds for single chain bias and MSE from Theorem 2 and Theorem 3, respectively, we have

$$\leq \sum_{s=1}^{S} \frac{T_s^2}{T^2} D_2' \left( \frac{1}{T_s} + \left( \tilde{M}_1 \tau^2 h_s^2 + \tilde{M}_2 h_s^{2K} \right) \right)$$

$$+ \sum_{i \neq j} \frac{T_i T_j}{T^2} D_1 \left( \frac{1}{T_i} + \left( M_1 \tau h_i + M_2 h_i^K \right) \right) D_1 \left( \frac{1}{T_j} + \left( M_1 \tau h_j + M_2 h_j^K \right) \right)$$

$$\leq D_2 \left( \frac{1}{T} + \frac{S^2 - S}{T^2} + \sum_{i,j} \frac{T_i T_j}{T^2} \left( M_1^2 \tau^2 h_m^2 + M_2^2 h_m^{2K} \right) \right)$$

$$\leq D_2 \left( \frac{1}{T} + \frac{S^2 - S}{T^2} + \frac{S^2 T_m^2}{T^2} \left( M_1^2 \tau^2 h_m^2 + M_2^2 h_m^{2K} \right) \right) ,$$

where $D_2 = \max\{D_2', D_1^2\}$, $T_m \triangleq \max_l T_l$, $h_m \triangleq \max_l h_l$, the last equality collects the low order terms. This completes the proof.

## H    Proof of Theorem 7

*Proof.* Following the proof of Theorem 6, for the variance, we have

$$
\mathbb{E}\left(\hat{\phi}_L^S - \mathbb{E}\hat{\phi}\right)^2 = \mathbb{E}\left(\sum_{s=1}^{S}\frac{T_s}{T}\left(\hat{\phi}_{L_s} - \mathbb{E}\hat{\phi}_{L_s}\right)\right)^2
$$

$$
= \sum_{s=1}^{S}\frac{T_s^2}{T^2}\mathbb{E}\left(\hat{\phi}_{L_s} - \bar{\phi}_{L_s}\right)^2 + \sum_{i\neq j}\frac{T_iT_j}{T_2}\mathbb{E}\left[\hat{\phi}_{L_i} - \mathbb{E}\hat{\phi}_{L_i}\right]\mathbb{E}\left[\hat{\phi}_{L_j} - \mathbb{E}\hat{\phi}_{L_j}\right]
$$

$$
= \sum_{s=1}^{S}\frac{T_s^2}{T^2}\mathbb{E}\left(\hat{\phi}_{L_s} - \bar{\phi}_{L_s}\right)^2 .
$$

Substituting the variance bound in Theorem 4 for each server, we have

$$
\mathbb{E}\left(\hat{\phi}_L^S - \mathbb{E}\hat{\phi}\right)^2 \leq D\sum_{s=1}^{S}\frac{T_s^2}{T^2}\left(\frac{1}{L_sh_s} + h_s^{2K}\right)
$$

$$
= D\sum_{s=1}^{S}\left(\frac{T_s}{T^2} + \frac{T_s^2}{T^2}h_s^{2K}\right)
$$

$$
= D\left(\frac{1}{T} + \sum_{s=1}^{S}\frac{T_s^2}{T^2}h_s^{2K}\right)
$$

$\square$

## I    Additional Results

See Figure 6 7 8 9 10 11. The content of the figures is described in the titles.

Figure 6: Testing loss vs. #workers. From top down, each row corresponds to the a9a, MNIST and CIFAR dataset, respectively.

Figure 7: Testing loss vs. #servers. From left to right, the first row corresponds to the a9a, MNIST datasets, and the second row corresponds to the CIFAR dataset, respectively.

Figure 8: Testing loss vs. #servers. From top down, each row corresponds to the a9a, MNIST and CIFAR dataset, respectively. Each server is associated with 1 worker.

Figure 9: Testing loss vs. #servers. From top down, each row corresponds to the a9a, MNIST and CIFAR dataset, respectively. Each server is associated with 2 workers.

Figure 10: Testing loss vs. #servers. From top down, each row corresponds to the a9a, MNIST and CIFAR dataset, respectively. Each server is associated with 4 workers.

Figure 11: Testing loss vs. #servers. From top down, each row corresponds to the a9a, MNIST and CIFAR dataset, respectively. Each server is associated with 6 workers.