[Reviews · NeurIPS 2016]

Reviewer 1

Summary

The paper develops theory for the stale stochastic gradient MCMC algorithm which can be useful to develop distributed stochastic gradient MCMC algorithms. The theory tells that although the bias and MSE are affected by the staleness of the stochastic gradient, the estimation variance is independent of the staleness. This is important because then the bound on the variance can linearly be reduced by the number of parallel workers. The authors also claim that the bias and MSE level can be maintained at the same level because in practice the effect of staleness can be canceled out by the increased speed of sampling. The theory is validated in the experiments on a synthetic dataset as well as Bayesian deep neural networks.

Qualitative Assessment

The paper is well written and I enjoyed it. Also, the theory seems to be useful to develop more efficient distributed stochastic gradient MCMC methods. It would have been more interesting if the effect of performing multiple updates per stale update (similar to the trajectory sampling in D-SGLD) can be studied.

Confidence in this Review

2-Confident (read it all; understood it all reasonably well)


Reviewer 2

Summary

This paper analyzes how SG-MCMC behaves when run asynchronously -- that is, when the gradients might be for older versions of the variables. The paper shows that the variance (after a fixed number of iterations) is not really harmed by the staleness, so the increased number of sample obtained by running asynchronously improves the overall variance.

Qualitative Assessment

Technical quality: I think that the theory is very complete (bounds are given for pretty much everything relevant to the problem), and the experiments show that this method performs better on large/complicated models (the small/simple models have too little variance for extra servers to help, and the staleness prevents much benefits). I think the biggest limitation of the paper is the lack of comparison against the method in [14] (the paper mostly compares against the non-distributed -- 1 worker -- case, instead of a more standard distributed case). Novelty/originality: My impression is theoretical results are mostly a combination of proof techniques used in other SG-MCMC and asynchronous SGD papers (however, I'm not too sure that this claim is correct). Assuming this is true, I think the results are well-executed, but not too unique. Potential impact or usefulness: I think the theoretical analysis will be useful for people interested in how asynchrony affects SG-MCMC. However, I'm not too clear how much this will help for running SG-MCMC in practice. My understanding is the following: - [14] gives a method of running on multiple workers synchronously (without stale gradients). - This paper shows that the stale gradients basically don't matter => you can just run asynchronously. However, it's not too clear to me how the method in this paper and the method in [14] compare with each other. As far as I can tell, [14] does not require too much synchronization, so it should run fairly quickly too. Clarity and presentation: I think that the paper was well-written and fairly easy to understand (I'm not very familiar with the area, and I could follow the paper without much trouble). I think that the biggest issue with clarity is that the paper doesn't make it clear how this result is related to [14]. In particular, I think that: - there should be just a quick comment at the beginning of section 3 or 3.1 saying that Appendix A gives one method of getting stale gradients (otherwise, based on reading [14], I think that it might appear like there's no reasonable system that generates stale gradients) - there should be some brief explanation of how this performance compares with running SG-MCMC in a distributed, but synchronous manner

Confidence in this Review

1-Less confident (might not have understood significant parts)


Reviewer 3

Summary

This paper presents theoretical analysis for Stochastic-Gradient MCMC methods using stale (outdated) gradients, which often appear in asynchronous settings. In a nutshell, the paper extends the results of Chen, Ding, Carin, NIPS 2015 to the stale gradient settings. The bias, MSE, and the variance of the SG-MCMC estimator is rigorously analyzed under certain and rather standard assumptions. It has been shown that the bias and the MSE of the estimator depend on the "staleness" of the gradients whereas the variance does not. These results indicate that in a distributed setting where multiple chains are run in parallel, the SG-MCMC estimator would be able to attain lower MSE even if stale gradients are being used. The theory is supported by several experiments (both synthetic and real-data).

Qualitative Assessment

Overall, in my opinion this is an excellent paper. The paper is very well written, the objectives and the contributions are stated very clearly. I really enjoyed reading the paper. Even though SG-MCMC methods have enabled scaling up MCMC to very large-scale settings, distributed and/or asynchronous variants of SG-MCMC have not been rigorously explored yet. In this sense, this paper has very important contributions and would be of great interest to the community. The results of the paper, in a sense, demystify asynchronous SG-MCMC, where for the first time, we obtain a clear understanding of the behaviors of the bias, variance, and the MSE of the SG-MCMC estimator. Besides, certain trade-offs become theoretically more grounded, such as the one explained in line 291 (page 7). Another take-home message is that using an higher-order integrator would not be useful in the stale-gradient settings. The results presented in the paper can also be used for developing and analyzing new distributed/asynchronous SG-MCMC methods, or even new stochastic optimization methods based on simulated annealing. I only have some minor comments/corrections. -- Main Document -- * Line 13: I think it is better to write "with respect to" instead of "w.r.t." since there is enough space. * It is quite trivial but it might be good to remind the reader about MSE = Bias^2 + Var. I would suggest the authors to check Section D of the supplementary document of (Li et al. 2016). ([28] in the paper) * Line 168: "M" should be "S". * Line 241: "indicates" should be "indicated" * Line 249: I would prefer saying "we average" since all the rest of the paragraph is in the present tense. * Theorem 6: Should the step length be "h_m" instead of "h"? * Line 291: I am not very happy with the usage of PSGLD as the baseline SG-MCMC method in the deep learning experiments. Due to the negligence of the "correction term" in PSGLD, the bounds given in Lemma 1 no longer hold for PSGLD: there are additional terms appearing in the bounds of both bias and MSE due to this negligence. Therefore, the theory presented in this paper is not completely appropriate for PSGLD. I would suggest the authors to use another SG-MCMC method which would obey the bounds given in Lemma 1. If this is not possible I would recommend providing a discussion/explanation on this subject. * Lines 322-325: The full list of authors should be given in [7,8,9]. -- Supplementary Document -- * Line 427, Line 448, Line 455: $X$ should be $x$ (lower-case). * The first equation after line 458: There should not be an absolute value operator on the right hand side. If there is, then it should be \leq (after applying a triangle inequality) * The last equation after line 458: I do not really understand how ||E \nabla \psi|| becomes ||\nabla \psi||. \nabla \psi is not deterministic, how come it is equal to its expectation? * Eq 19: (and the equation after that) h should be h_s. Eq 20: h should be h_m. The same problem appears in the equations after Line 511. This should be also corrected in all the relevant parts of the main text.

Confidence in this Review

3-Expert (read the paper in detail, know the area, quite certain of my opinion)


Reviewer 4

Summary

In this paper, the authors propose a variant of the Stochastic Gradient MCMC algorithm which uses state stochastic gradient instead of current gradient. A thorough theory has been provided to bound the bias and MSE. More importantly, this method is applied to distributed asynchoronous system.

Qualitative Assessment

The idea of this paper is simple, interesting and compatible with distributed system and computed asynchronously. Thorough theoretical analysis guarantees that the bias and MSE can be bounded. With regard to empirical results, the authors evaluate its effectiveness on both read and simulated data. However, there exist some rooms to improve. Firstly, I don't think the average iterations is a good metric. Directly comparing the running time seems better. Secondly, from the figure (mainly from supplement) I observe that this method achieves significant improvements on a9a dataset, but improves slightly in the other two datasets in terms of clock time. Thirdly, in deep learning experiment, Preconditioned SGLD is employed as the baseline method, which I think is not the mainstream SG-MCMC method. I suggest the authors to choose some classical methods, say, SGLD or SGHMC as baseline methods.

Confidence in this Review

2-Confident (read it all; understood it all reasonably well)


Reviewer 5

Summary

This paper presents an analysis (both theoretical and empirical) on distributed stochastic gradient MCMC algorithms under stale gradients. In this setting, which is commonly found in asynchronous distributed systems, each worker performs gradient-based updates using an older ("stale") version of the gradient computed in a previous iteration. Given a fixed bound on the staleness of any gradient (in any update), this paper provides theoretical results bounding three quantities---the bias, MSE, and variance---involving the expectation of a test function with respect to a posterior estimate computed via stale gradients. Empirically, this paper shows results for a toy Gaussian model on synthetic data (demonstrating convergence accuracy), a Bayesian logistic regression model on synthetic data (showing variance reduction of the posterior estimate), and a Bayesian neural network on image classification data (demonstrating test loss vs time and iteration).

Qualitative Assessment

Comments: * Algorithmically, I do not feel that there is a great deal of novelty in this paper; the concept of stale gradients has been explored in previous literature (in an optimization setting), and so has the procedure for distributed stochastic gradient MCMC methods. Therefore, I feel that the main contribution of this paper is in the theoretical work showing bounds on the estimated posterior expectation under these methods. * The primary theoretical take-away in this paper seems to be that the estimation variance of the posterior estimate is independent of the staleness (and that this is not true for the bias and MSE of the estimate). However, I feel there is not enough clear discussion about how reduction of the variance allows for specific benefits in practice. The theory indicates that the bias and MSE worsen with more staleness; intuitively, I can see how this has a practical detrimental effect, and would yield a worse posterior approximation given too much staleness. However, how specifically will a reduction of variance allow for better posterior approximations? Perhaps it means that a lot of staleness will yield a stable, though incorrect estimate (i.e. an estimate with high MSE but low variance). And if so, what are the benefits of this in practice? It would be nice to have more discussion of this sort, that better connects the theoretical results with practical benefits. * I feel that the clarity of the writing in this paper was quite good. While this was a theoretically-focused paper, the authors do a good job of motivating the theory and making the notation and results clearly stated. However, as I describe in the point above, I feel there could be more done to provide accessible take-aways from the theoretical results. * The theoretical results are given in terms of the posterior average of a test function \phi. There are no conditions given in this paper on this test function, so I wanted to clarify: can this test function be any function of posterior parameters? Or is it assumed to be from a specific class of functions?

Confidence in this Review

2-Confident (read it all; understood it all reasonably well)